# Human Dendritic Cells Transmit Enterovirus A71 via Heparan Sulfates to Target Cells Independent of Viral Replication

Leanne C. Helgers,[a,b] Michel S. Bhoekhan,[a,b] Dasja Pajkrt,[c,d] Katja C. Wolthers,[c,d] (ID) Teunis B. H. Geijtenbeek,[a,b] (ID) Adithya Sridhar[b,c,d]

[a]Department of Experimental Immunology, Amsterdam UMC, University of Amsterdam, Amsterdam, the Netherlands
[b]Amsterdam Institute for Infection and Immunity, Amsterdam, the Netherlands
[c]Emma Children's Hospital, Department of Pediatric Infectious Diseases, Amsterdam UMC, University of Amsterdam, Amsterdam, the Netherlands
[d]Department of Medical Microbiology, OrganoVIR Labs, Amsterdam UMC, University of Amsterdam, Amsterdam, the Netherlands

Leanne C. Helgers and Michel S. Bhoekhan contributed equally to this work. Author order was determined at random.
Teunis B. H. Geijtenbeek and Adithya Sridhar also contributed equally to this work.

**ABSTRACT** Enterovirus A71 (EV-A71) is a causative agent of life-threatening neurological diseases in young children. EV-A71 is highly infectious but it remains unclear how the virus disseminates from primary entry sites—the mucosa of the respiratory tract or the intestine—to secondary replication sites—skin or brain. Here, we investigated the role of dendritic cells (DCs) in EV-A71 dissemination. DCs reside in the mucosa of the airway and gut, and migrate to lymphoid tissues upon activation and, therefore, could facilitate EV-A71 dissemination to secondary replication sites. Monocyte-derived DCs were not permissive to different genotypes of EV-A71 but, notably, coculture with EV-A71-susceptiblle RD99 cells led to very efficient infection of RD99 cells. Notably, EV-A71 transmission of DCs to RD99 was independent of viral replication as a replication inhibitor did not affect transmission. Soluble heparin blocked EV-A71 transmission by DCs to RD99 cells, in contrast to antibodies against known attachment receptor DC-SIGN. These results strongly suggest that DCs might be a first target for EV-A71 and involved in viral dissemination via heparan sulfates and heparin derivatives might be an effective treatment to attenuate dissemination.

**IMPORTANCE** EV-A71 is an emerging neurotropic virus that is of emerging concern and can result in polio-like illness. The exact mechanism of how EV-A71 results in neurological symptoms are unknown. In particular, the early dissemination of the virus from primary replication sites (airway and intestine) to secondary sites (central nervous system and skin) needs to be elucidated. There is evidence pointing toward a role for dendritic cells (DC) in EV-A71 transmission. Moreover, heparan sulfate (HS) binding mutations are observed in patients with severe diseases. Therefore, we evaluated the potential role of HS on DC in transmission. We find that HS are critical for transmitting EV-A71 by DC to target cells. Our data are consistent with other clinical and *in vitro* observations highlighting the importance of HS in EV-A71-induced disease.

**KEYWORDS** dendritic cells, enterovirus, picornavirus, receptors

Address correspondence to Teunis B. H. Geijtenbeek, t.b.geijtenbeek@amc.uva.nl, or Adithya Sridhar, a.sridhar@amsterdamumc.nl.

The authors declare no conflict of interest.

Enterovirus 71 (EV-A71) belongs to the *Picornaviridae* family and is the most prevalent causative agent of hand, foot, and mouth disease (HFMD) in young children (1–3). In a small percentage of infected children, EV-A71 causes life-threatening neurological disease because of transmission from the primary replication sites (respiratory or intestinal mucosa) to secondary target tissues, including the central nervous system (CNS) (4, 5). Consequently, EV-A71 infection may cause severe polio-like acute paralysis, meningitis, and encephalitis (1–5). At present, there is no antiviral treatment available and only genotype specific vaccines are available in China (2, 3). The primary entry sites for EV-A71 are via the mucosa of the respiratory tract or the intestine. However, it is not known how EV-A71

disseminates from these primary sites to secondary sites such as the brain and skin (4, 6). A potential mechanism is through the interaction of EV-A71 with immune cells at the primary sites resulting in dissemination into lymphoid tissue and subsequent secondary sites.

Dendritic cells (DCs) are antigen-presenting cells that reside in mucosal tissues and are, therefore, among the first immune cells to encounter pathogens upon infection (7, 8). DCs are essential for inducing immunity to invading pathogens as DCs capture and process pathogens for antigen presentation to naive T cells (9, 10). In turn, T cells induce a pathogen-tailored adaptive immune response. However, multiple studies have shown that the function of DCs is subverted by several viruses, e.g., measles virus, hepatitis C virus (HCV), dengue virus (DENV), and human immunodeficiency virus (HIV), resulting in viral dissemination mediated by DCs (11–14). In the case of EV-A71, the role DCs play in viral pathogenesis remains unclear. EV-A71 has been shown to productively infect immature DCs, whereas DCs also mediate viral *trans*-infection via the attachment receptor DC-SIGN independent of replication in DCs (15, 16). Furthermore, the latter study implicated a role for DC-SIGN in capturing and transferring the virus to target cells (16). Moreover, there is clinical data to indicate that heparan sulfate (HS) proteoglycans binding by EV-A71 may play a role in viral dissemination to secondary sites; however, the relevance of HS binding to DC interaction remains to be elucidated (17, 18).

Therefore, in this study, we investigated the role of DCs in EV-A71 transmission and potential surface molecules, including HS, that maybe involved in DC/EV-A71 interaction. Monocyte-derived DCs were not permissive to infection with EV-A71 but efficiently transmitted EV-A71 to target cells independent of viral replication in DCs. Soluble heparin blocked EV-A71 transmission by DCs. These data support a role for DCs in viral transmission and highlight the importance of HS in EV-A71 pathogenesis. Further research might benefit the search for new strategies for combating EV-A71 infection.

## RESULTS

**Human dendritic cells transmit Enterovirus A71 genotypes C1 and C5 to target cells.** We investigated whether DCs can transmit EV-A71 to target cells. DCs were exposed to EV-A71 C1 or EV-A71 C5 for 2 days and, after extensive washing, cocultured with RD99 cells for 2 days. Next, EV-A71 C1 and EV-A71 C5 transmission was determined by flow cytometry (Fig. 1A and 2A). DCs express high levels of CD1a (Fig. S1A), and CD1a expression was used to distinguish the DC population from RD99 cells. In turn, antibodies targeting EV-A71 were used to determine the number of EV-A71 positive cells. EV-A71 C1 and EV-A71 C5 positive RD99 cells were observed after 24-h posttransmission (p.t.) while the number of EV-A71 C1 and EV-A71 C5 positive RD99 cells significantly increased at 48 h p.t. (Fig. 1B and 2B), indicating transmission of EV-A71 from DCs to RD99. Strikingly, preincubation of DCs with ribavirin, a known inhibitor of EV-A71 (19, 20), did not block transmission, suggesting that viral transmission to RD99 cells is independent of EV-A71 replication in DCs (Fig. 1B and 2B). Ribavirin functionality was determined in RD99 cells (Fig. S1B and C). Next, the total number of EV-A71 RNA copies was determined in the supernatant of DCs coculture by quantitative PCR. EV-A71 C1 and C5 RNA copies increased over all time points and treatment of DCs with ribavirin did not affect virus production (Fig. 1C and 2C). Lastly, EV-A71 transmission was determined by assessing the cytopathic effect (CPE) by microscopy. Coculture of RD99 cells with EV-A71-treated DCs induced a strong CPE in contrast to mock infection (Fig. 1D and 2D). Furthermore, an equivalent CPE was observed when ribavirin was used to block infection of DCs. Altogether, these data strongly suggest that DCs were able to transmit different genotypes of EV-A71 to target cells independent of viral replication in DCs.

**Human dendritic cells are not permissive to EV-A71 infection.** Next, we investigated whether EV-A71 C1 and EV-A71 C5 could be detected in DCs. Importantly, no infection of EV-A71 in DCs was observed for both genotypes at 48 h postinfection (p.i.) (Fig. 3A and B). Moreover, coculture of DCs with RD99 cells did not lead to infection of DCs at 24 h p.t (72 h p.i.; Fig. 3A and B). In addition, direct infection of DCs alone did not result in infection by EV-A71 C1 and EV-A71 C5 as determined by flow cytometry and quantitative PCR (Fig. S2). These results strongly suggest that DCs are not susceptible to EV-A71 infection

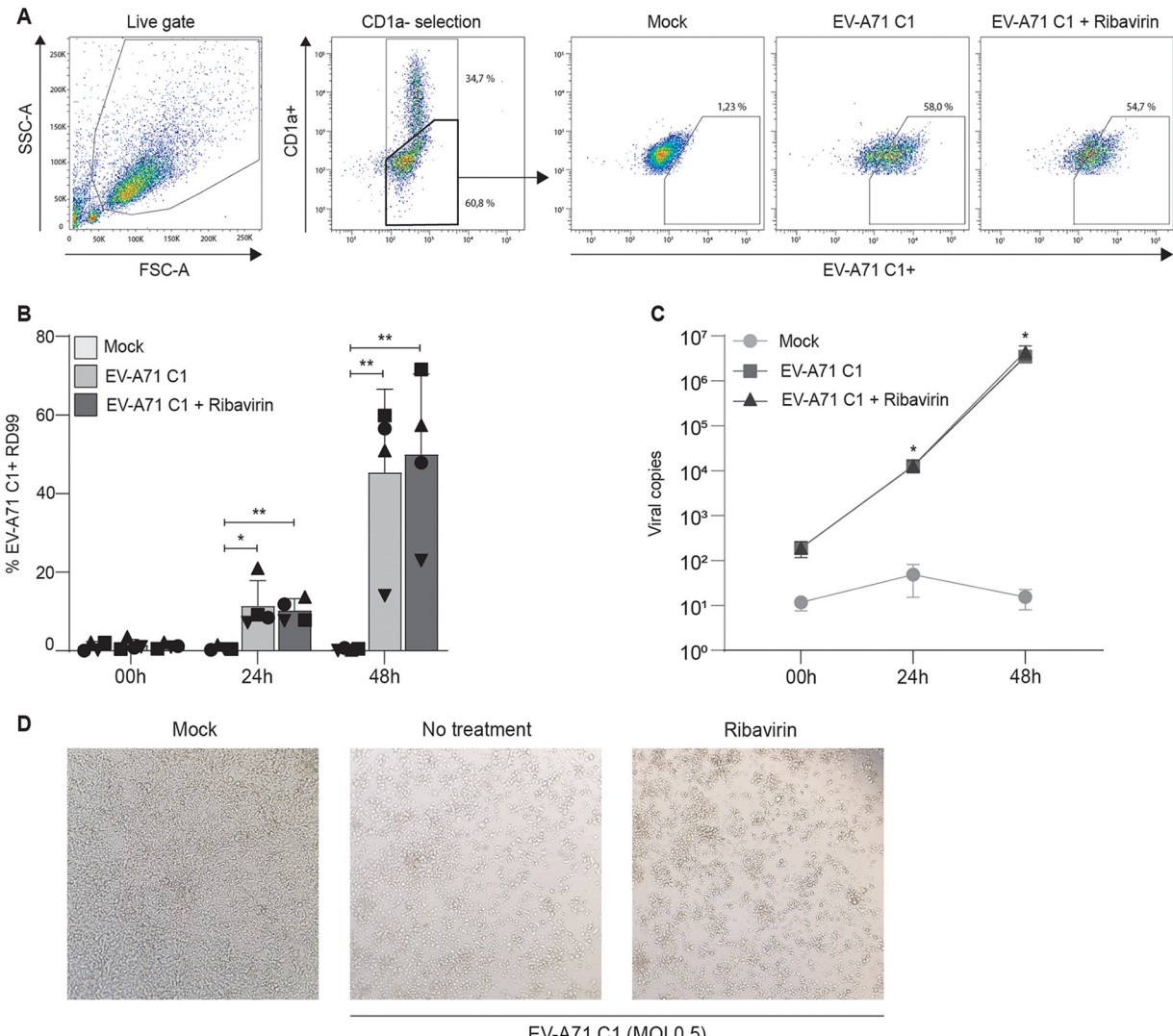

**FIG 1** Human dendritic cells transmit EV-A71 C1 to target cells. (A to D) DCs were exposed to EV-A71 C1 (MOI 0.5) for 2 days, and after extensive washing cocultured with RD99 cells. DCs were treated with Ribavirin (RBV; 10 $\mu$M) to control for a replicating infection in DCs during transmission. After coculture, supernatant and cells were collected at different time points and analyzed by RT-qPCR and flow cytometry, respectively. (A to C) Flow cytometry analyses of DC transmission to RD999 is shown for one representative donor (A) and combined data for different donors on infection of CD1a-negative RD99 cells by flow cytometry (B, N = 4) and virus production in coculture (C, N = 4). The symbols represent independent donors mean ± s.d. (flow cytometry) or mean ± SEM (RT-qPCR) of duplicates. (D) Images show the morphological changes of RD99 cells indicating the cytopathic effects 48 h after incubation with a representative DC donor.

following early encounters with the virus but are still able to efficiently transmit the virus to target cells.

**Heparan sulfates are involved in EV-A71 transmission by DCs.** Next, we investigated the receptors involved in virus transmission by DCs using blocking antibodies against several putative viral receptors. Several receptors already have been implicated in EV-A71 infection and transmission of DCs, including DC-SIGN (15, 16, 21–23). In addition, HS are implicated in EV-A71 infection, as potential attachment receptors (17, 18). HS are expressed in DCs, and we used soluble heparin to block HS bindings sites on EV-A71. Antibodies against DC-SIGN did not block EV-A71 transmission by DCs. Strikingly, preincubation of EV-A71 C1 and EV-A71 C5 with soluble heparin abrogated transmission of both strains at 24 h and 48 h to background levels (Fig. 4B and 5B). Similarly, EV-A71 RNA copies in the supernatant increased over time in DC-RD99 cocultures but soluble heparin significantly decreased EV-A71 copies compared with the untreated control (Fig. 4C and 5C). These results strongly suggest that HS are involved transmission of both EV-A71 C1 and C5 genotypes by DCs.

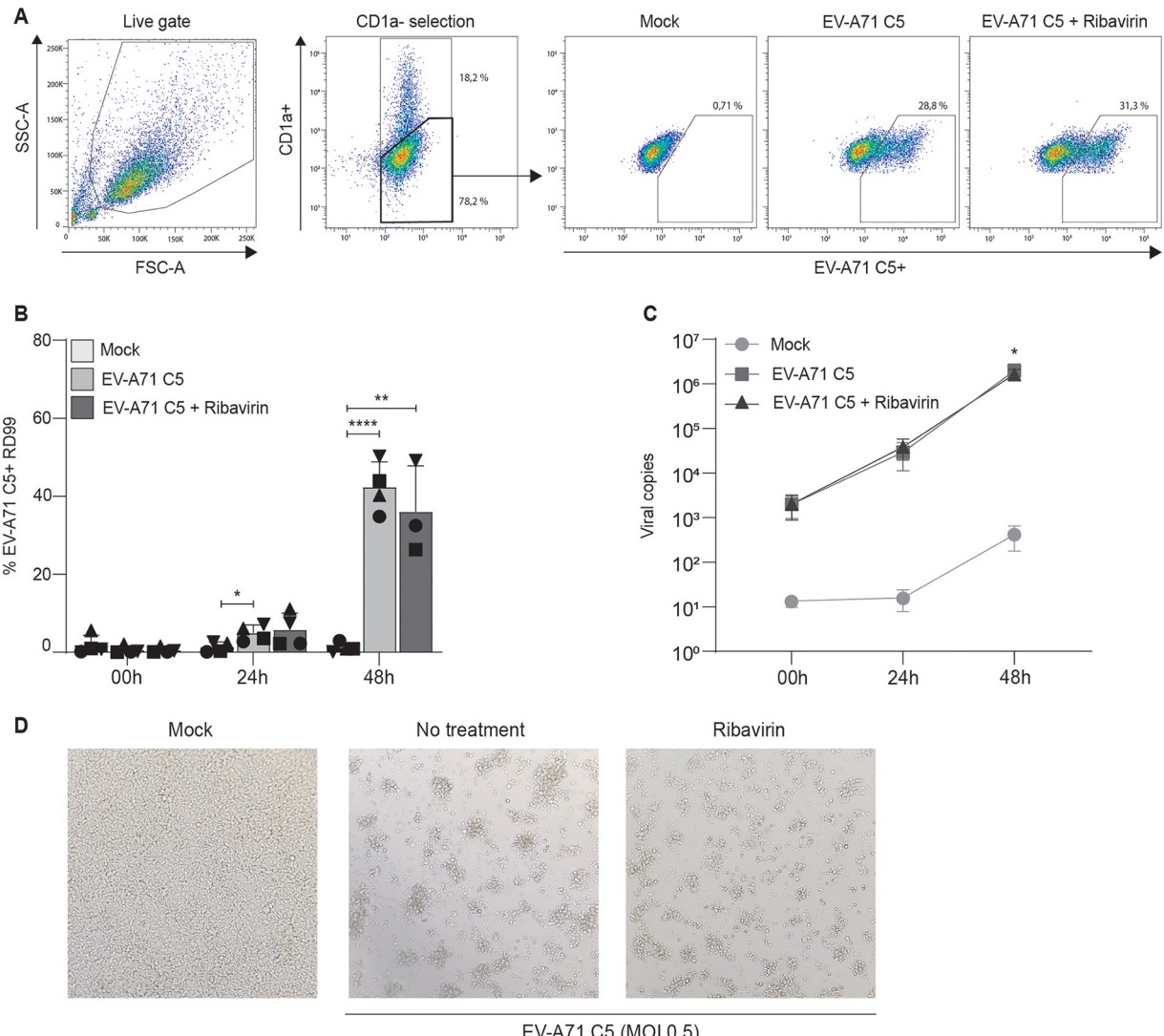

**FIG 2** Human dendritic cells transmit EV-A71 C5 to target cells. (A to D) Transmission experiments with EV-A71 C5 are performed as described in Fig. 1. (A) Populations plots of one representative donor are shown to indicate the percentages of gated cells, (B, N = 4) followed by the combined percentage of EV-A71 VP1 positive RD99 cell (CD1a-negative population) as detected by flow cytometry and (C, N = 4) the total amount of viral copies as detected by RT-qPCR. The symbols represent independent donors mean ± s.d. (flow cytometry) or mean ± SEM (RT-qPCR) of duplicates. (D) Images show the morphological changes of RD99 cells indicating the cytopathic effects 48h after incubation with a representative DC donor.

## DISCUSSION

Our data strongly suggest that DCs are not susceptible to EV-A71 infection but can efficiently transmit captured virus to target cells. Previous research by Ren et al. showed *trans*-transmission of EV-A71 to target cells by DCs using short (4 h) DC/EV-A71 incubation periods (16). However, this short duration is not sufficient to assess *cis*-transmission, therefore, warranting longer incubation periods such as the 48 h used in our study. Exposing the DCs to EV-A71 in the presence of viral replication inhibitor ribavirin for this longer duration still resulted in transmission of the virus to target cells. Additionally, we showed that no EV-A71 could be detected in DCs up to 72 h p.i. (24 h p.t.), confirming that DCs are not permissive to infection. This was confirmed by the lack of EV-A71 replication in DCs alone when exposed to EV-A71 C1 and C5. This is in contrast to the study of Lin et al. who report active replication of EV-A71 in DCs. However, the viral titer in the study by Lin et al. declined steadily over 48 h suggesting that DCs retain EV-A71 but active infection was not conclusive (15).

Next, to assess the potential receptors involved in the *trans*-transmission of EV-A71 by DCs, we evaluated the role of DC-SIGN and HS. DC-SIGN has been shown to be involved

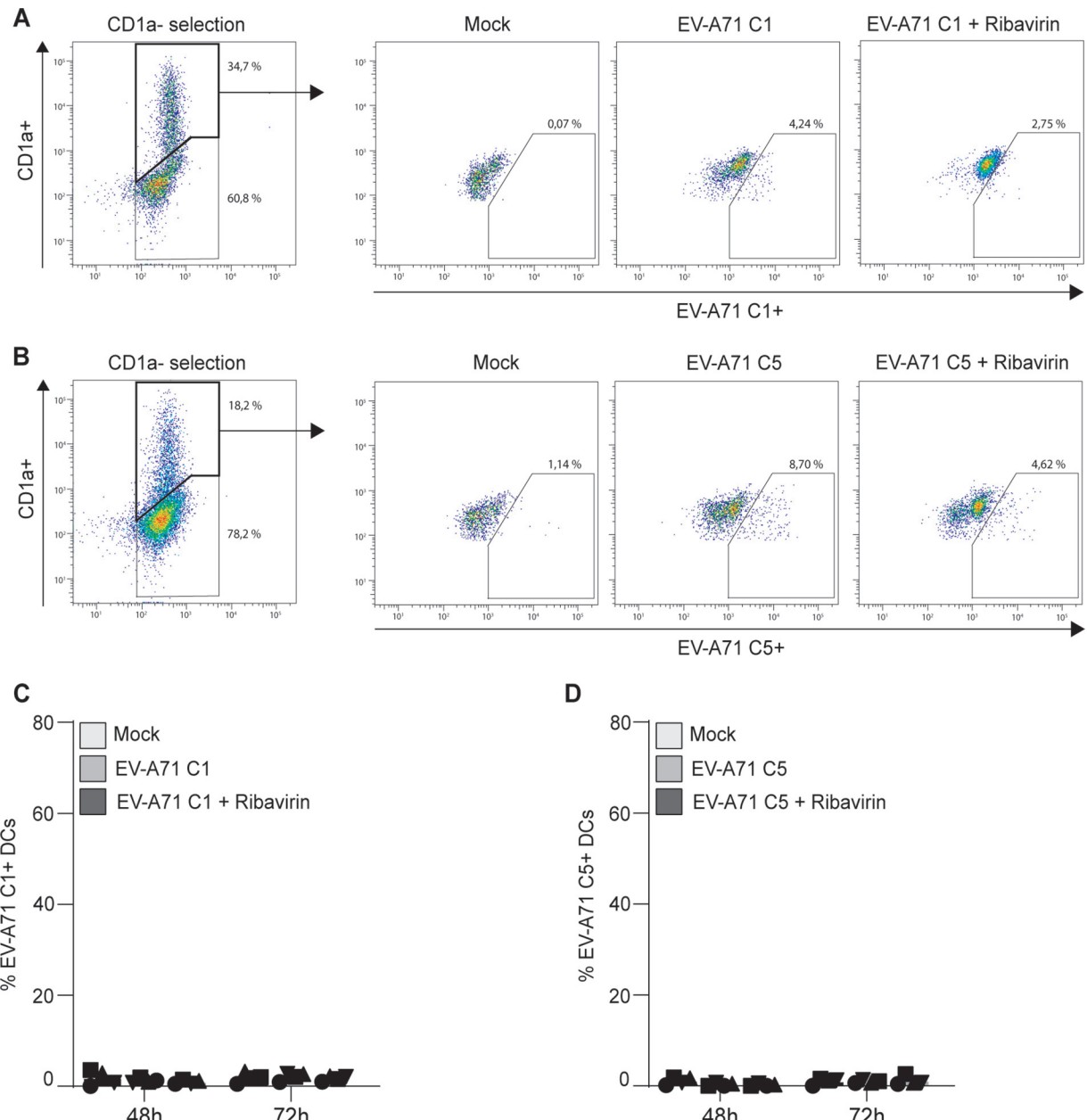

**FIG 3** Human dendritic cells transmit EV-A71 C1 to target cells. (A to D) DCs were exposed to EV-A71 C1 or EV-A71 C5 (MOI 0.5) for 2 days in the absence or presence of replication inhibitor ribavirin (10 μM). Next, DCs were washed extensively and cocultured with RD99 cells. DCs from Fig. 1 and Fig. 2 were collected and analyzed by flow cytometry, respectively, at the moment of coculture (48 h) or 1 day (72 h) after transmission. (A) Populations plots of one representative DC donor are shown to indicate the percentages of gated cells followed by the combined percentage of (B, N = 4) EV-A71 C1 and (C, N = 3) EV-A71 C5 in the DC population (CD1a-positive population) as determined by flow cytometry. The symbols represent independent donors mean ± s.d. of duplicates.

for *trans*-transmission of EV-A71 by DCs. However, in our study, treatment of DCs with anti-DC-SIGN antibodies did not result in inhibition. The differences in the finding could be due to the different EV-A71 strains or differences in exposure times used in studies. Clinical studies have shown an important role for DC-SIGN in disease outcome as polymorphisms in DC-SIGN, and high levels of soluble DC-SIGN are associated with more severe HFMD outcome as the virus is less efficiently cleared (7, 8).

Interestingly, our results show a complete abrogation of EV-A71 transmission when the virus is preincubated with soluble heparin both in flow cytometry and RT-qPCR results. As HS is seen as primarily an attachment, rather than entry receptor of EV-A71,

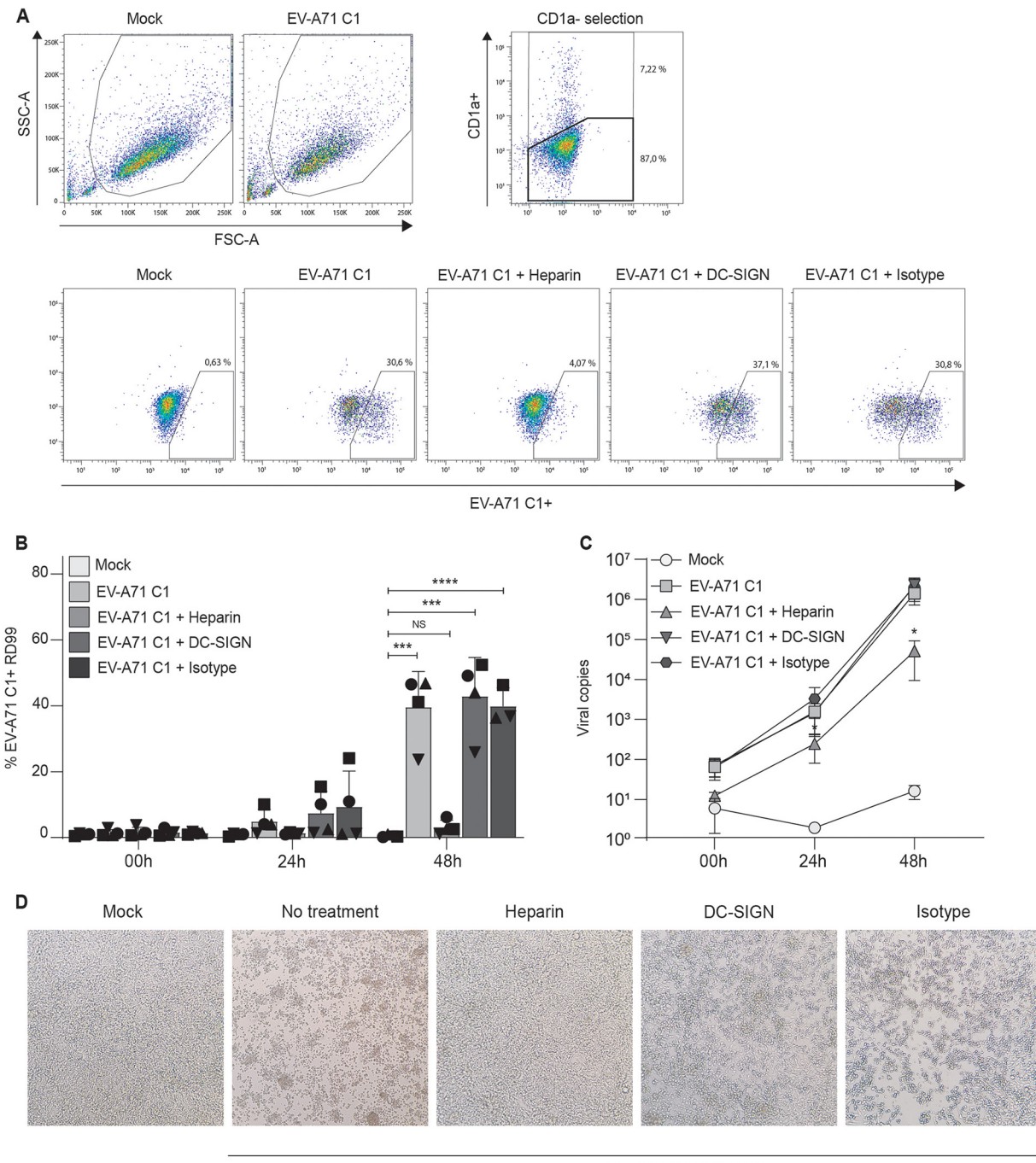

**FIG 4** Heparan sulfates are involved in EV-A71 C1 transmission by DCs. (A to D) Transmission experiments with EV-A71 C1 are performed as described in Fig. 1. DCs were preincubated with either anti-DC-SIGN antibodies or soluble heparin (5 mg/mL). After coculture, supernatant and cells were collected at different time points, and analyzed by RT-qPCR and flow cytometry, respectively (A) Populations plots of one representative DC donor are shown to indicate the percentages of gated cells, (B, N = 4) followed by the combined percentage of EV-A71 C1 positive cells RD99 cells (CD1a-negative population) as determined by flow cytometry, (C, N = 4) and the total amount of viral copies as detected by real-time PCR. The symbols represent independent donors mean ± s.d. (flow cytometry) or mean ± SEM (RT-qPCR) of duplicates. (D) Images are included to show the morphological changes of RD99 cells 48 h after incubation with a representative DC donor.

the role of HS in EV-A71 transmission may be more prominent compared to earlier studies with PSGL-1 or DC-SIGN. HS-binding mutations in VP-1 of EV-A71 appear to be important for neurotropism of EV-A71 (24). HS-binding mutations such as glutamine at 145Q are associated with severe neurological diseases in human. Moreover, Tseligka et al. showed intrahost HS-binding adaptations leading to HS-binding variants being isolated

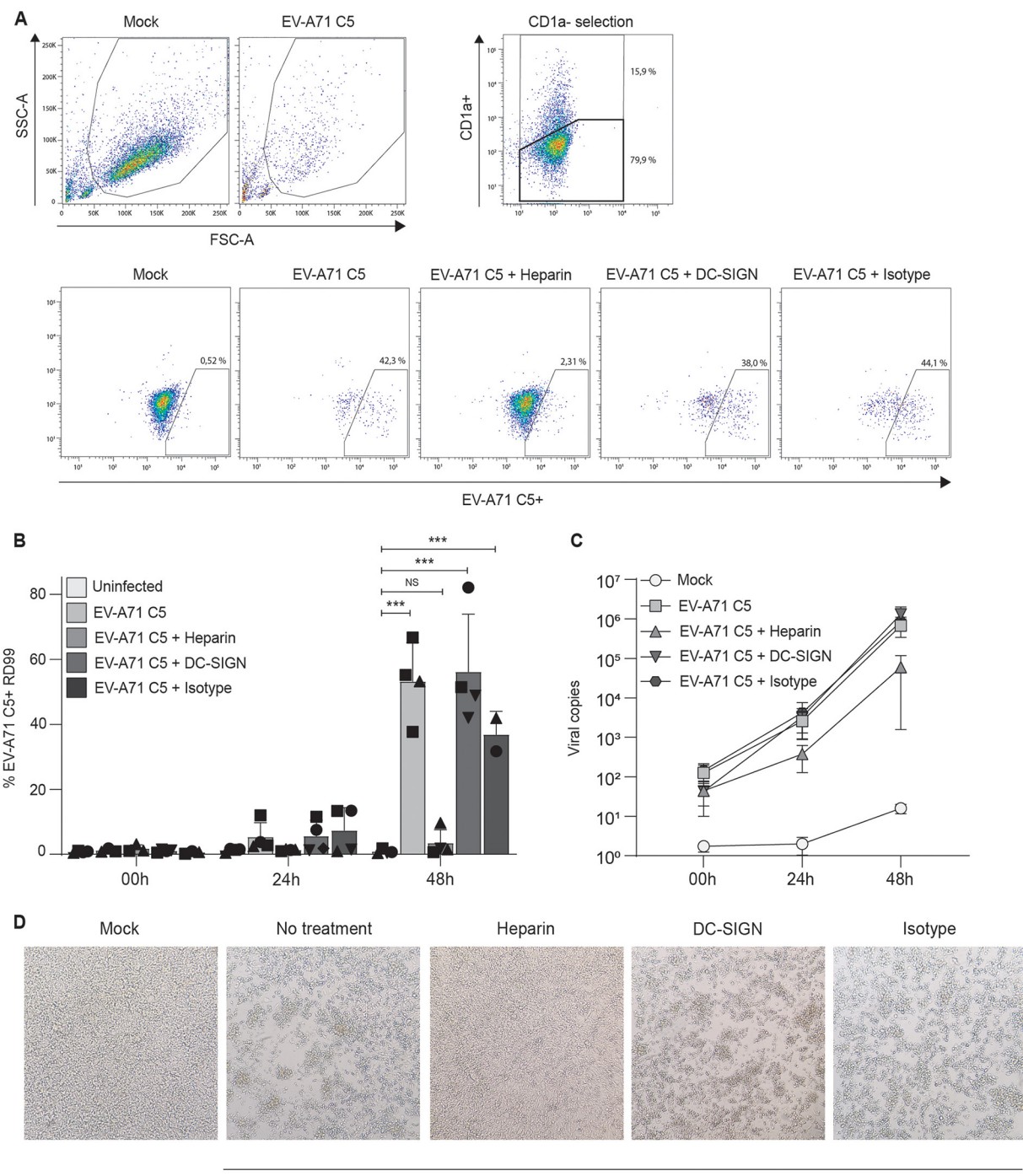

**FIG 5** Heparan sulfates are involved in EV-A71 C5 transmission by DCs. (A to D) Transmission experiments with EV-A71 C5 are performed as described in Fig. 4. (A) Populations plots of one representative DC donor are shown to indicate the percentages of gated cells, (B, N = 4) followed by the combined percentage of EV-A71 C5 positive cells RD99 cells (CD1a-negative population) as determined by flow cytometry, (C, N = 4) and the total amount of viral copies as detected by real-time PCR. The symbols represent independent donors mean ± s.d. (flow cytometry) or mean ± SEM (RT-qPCR) of duplicates. (D) Images are included to show the morphological changes of RD99 cells 48h after incubation with a representative DC donor.

from secondary infection sites (24). Thus, interaction of HS-binding variants with DCs may play a role in their dissemination to secondary target sites. Comparison of HS-binding and non-HS-binding variants as well as their transmission by DCs to susceptible models such as brain organoids will be of importance to further elucidate this dependence. Most importantly, should these findings hold true, low molecular weight heparin could be an option to

prevent dissemination of virus to secondary sites in infected infants. With the lack of an effective treatment or vaccine, unravelling this mechanism might prove valuable in combating EV-A71 related pathologies.

## METHODS

**Monocyte-derived dendritic cells.** Peripheral blood monocytes were isolated from buffy coats of healthy donors (Sanquin) by Lymphoprep (Axis-Shield) gradient, followed by Percoll (Amersham Biosciences) gradient steps. Monocytes were differentiated into monocyte-derived DCs in the presence of 500 U/mL IL-4 (Invitrogen) and 800 U/mL GM-CSF (Invitrogen) for 6 days in RPMI supplemented with 10% fetal calf serum (FCS) (Invitrogen), 10 U/mL penicillin (Invitrogen), 10 mg/mL streptomycin (Invitrogen), and 2 mM L-glutamine (Lonza) as described before (25). Dendritic cells were routinely analyzed for surface expression of DC-markers CD1a and DC-SIGN, and activation markers CD80, CD83, and CD86 (Fig. S1).

**Enterovirus A71.** EV-A71 AB552982 was obtained from the National Institute for Public Health and the Environment, Bilthoven, and strain KU697336 was previously provided by K. Mizuta (Yamagata Prefectural Institute of Public Health, Japan), N.T.T. Thao, and P.V. Tu (Pasteur Institute Ho Chi Minh City, Vietnam) (26). EV-A71 (C1 and C5) were propagated in RD99 cells. Supernatant was filtrated, stored at $-80°C$, and thawed to room temperature (RT) before use. Infectious virus titers of the strains were determined by the micro titration method on RD99 and were expressed as the 50% tissue culture infectious dose (TCID50) (27).

**Cell line continuation.** RD99 cells were passaged weekly, after which cells were maintained in Eagle's minimum essential medium (EMEM; Lonza) supplemented with 8% fetal bovine serum (FBS, Lonzo), Pen-Strep (10,000 U/mL each; Lonza), $100\times$ non-essential amino acids (NEAA; Sciencell), and L-glutamine (200 mM; Lonza) in 0,85% NaCl.

**Transmission.** DCs were exposed to EV-A71 C1 and EV-A71 C5 at MOI 0.5 for 48 h at 37°C either without the presence of or in the presence of (1 h at 37°C preincubation) replication inhibitor ribavirin (10 $\mu$M), anti-DC-SIGN (20 $\mu$g/$\mu$L), or an isotopic IgG1-kappa control (E-bioscience; 20 $\mu$g/$\mu$L). In addition, EV-A71 C1 and EV-A71 C5 were preincubated with 5 mg/mL soluble heparin at 37°C for 2 h for HS proteoglycan binding assay. After 48 h, DCs were washed extensively to remove unbound virus and cocultured with RD99 cells for 48 h at 37°C. After coculture, supernatant was collected and cells were fixed in 4% para-formaldehyde at different time points (0 h, 24 h, 48 h). Supernatant was analyzed using quantitative real-time PCR (RT-qPCR) whereas cells were analyzed using flow cytometry.

**Flow cytometry.** After fixation, cells were permeabilized in phosphate buffered saline (PBS) supplemented with 0.5% saponin and 0.5% bovine serum albumin (BSA) for 10 min. Cells were stained with anti-EV-A71 (1:800, Geno) followed by Alexa 488-conjugated Donkey-anti-rabbit (1:200, Invitrogen) in combination with APC-conjugated CD1a (1:25, BD Biosciences). Cells were quantified on a FACS Canto II (BD Biosciences) and results were analyzed using FlowJo software.

**RNA isolation, cDNA synthesis, and real-time PCR.** RNA from collected samples was extracted according to the manufacturer's protocol in the isolate II RNA minikit (Bioline). After RNA isolation, the eluates were reverse transcribed into cDNA using 20 mg/mL $\alpha$-casein, $10\times$ CMB1-buffer (pH 8.3), 40 mM dNTP's (10 mM each; Roche), 1.5 $\mu$g/$\mu$L primer random hexamers, 100 mM MgCl$_2$,20 to 40 units/$\mu$L inhibitor RNAsin (Promega), and 200 units/$\mu$L SuperScript II (RNase H negative; Thermo Fisher Scientific). The reverse transcription reactions were performed for 30 min at 42°C at 600 rpm on a heat block. Before running the RT-qPCR (CFX), the cDNA mixtures were stored at $-20°C$ and thawed prior to RT-qPCR. During RT-qPCR, cDNA mixture was amplified with primers for EV-A71 genome, 50 $\mu$M EV-A71 forward, 5'-GGC CCT GAA TGC GGC TAA T-3', and 50 $\mu$M EV-A71 reserve, 5'-GGG ATT GTC ACC ATA AGC C-3'. RT-qPCR was run using the Bio-Rad CFX system with $2\times$ SYBR green I Dye. RT-qPCR yielded Cq values and viral RNA copies were calculated with the equation derived from the standard curve for Enterovirus primers: $y = -3.3968x + 38.939$ (R2 = 1). Cq was substituted in place of y and solving x gives viral copy numbers.

**Statistical analysis.** Statistical analyses were performed using the independent T-test for unpaired observations using GraphPad Prism 8 software. Statistical significance was set at $P < 0.05$.

**Ethical statement.** This study was done in accordance with the ethical guidelines of the Academic Medical Center and human material was obtained in accordance with the AMC Medical Ethics Review Committee. Buffy coats obtained after blood donation (Sanquin) were handled anonymously.

## SUPPLEMENTAL MATERIAL

Supplemental material is available online only.

**SUPPLEMENTAL FILE 1**, PDF file, 0.7 MB.

## ACKNOWLEDGMENTS

This work was supported by the Netherlands Organization for Scientific Research (NWO) VIDI Grant 91718331, the European Research Council (ERC) Advanced Grant 670424, and the ZonMW Meer Kennis met Minder Dieren Grant 114021506.

L.C.H., M.S.B., and A.S. designed, performed, and interpreted most experiments and prepared the manuscript; L.C.H., M.S.B., D.P., K.C.W., T.H.B.G., and A.S. participated in discussion of the data; T.B.H.G. and A.S. supervised all aspects of the project.

We declare no competing interests.

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
