## [Reviewer comments · Microbiology Spectrum]

Microbiology Spectrum

Human dendritic cells transmit Enterovirus A71 via heparan sulfates to target cells independent of viral replication

Leanne Helgers, Michel Bhoekhan, Dasja Pajkrt, Katja Wolthers, Teunis Geijtenbeek, and Adithya Sridhar

Corresponding Author(s): Adithya Sridhar, Amsterdam UMC

Review Timeline:

Submission Date:	July 22, 2022
Editorial Decision:	August 12, 2022
Revision Received:	September 15, 2022
Accepted:	September 27, 2022

Editor: Vaithilingaraja Arumugaswami

Reviewer(s): The reviewers have opted to remain anonymous.

Transaction Report:

DOI: <https://doi.org/10.1128/spectrum.02822-22>

August 12, 2022

Dr. Adithya Sridhar
Amsterdam UMC
Medical Microbiology
Amsterdam
Netherlands

Re: Spectrum02822-22 (Human dendritic cells transmit Enterovirus A71 via heparan sulfates to target cells independent of viral replication)

Dear Dr. Adithya Sridhar:

Thank you for submitting your manuscript to Microbiology Spectrum. Now we have received comments from the reviewers. Though, this study is interesting, the reviewers' have raised some concerns on lack of proper controls and have provided additional suggestions. When submitting the revised version of your paper, please provide (1) point-by-point responses to the issues raised by the reviewers as file type "Response to Reviewers," not in your cover letter, and (2) a PDF file that indicates the changes from the original submission (by highlighting or underlining the changes) as file type "Marked Up Manuscript - For Review Only". Please use this link to submit your revised manuscript - we strongly recommend that you submit your paper within the next 60 days or reach out to me. Detailed instructions on submitting your revised paper are below.

Link Not Available

Sincerely,

Vaithilingaraja Arumugaswami

Journals Department
Reviewer comments:

Reviewer #1 (Comments for the Author):

Helgers et al. analyzed the potential role of DCs in EV-A71 dissemination. The experiments are well designed. I have some concerns about the technical issues.

1. Ribavirin could not suppress EV-A71 infection (see Smee et al., 2016, Antiviral Research). Guanidine HCl (2 to 4 mM) or rupintrivir (1 μ M) would be useful for the aim. Suppression of EV-A71 infection by these inhibitors must be confirmed to validate the experiments. Therefore, the potential effect of replication in DCs could not be excluded.
2. Protein secretion is inhibited in enterovirus-infected cells (Spagnolo and Kirkegaard, 2009, JVI). Therefore, a cell surface

marker CD1a might not serve as the marker for the EV-A71-infected DCs. To clarify the absence of EV-A71 infection in DCs, the authors should add the data of only DCs, thus in the absence of RD99 cells, in the context of experiments in Fig.1.

3. L136: CD1a?

Reviewer #2 (Comments for the Author):

Helgers et al present a study investigating the role of dendritic cells (DC's) in the transmission of EV-A71. Using a combination of flow cytometry and qRT-PCR the authors demonstrate that, while not susceptible to EV-A71 infection themselves, DC's are capable of transmitting infectious virus to susceptible cells following co-culture, and doing so in a replication-independent manner. They subsequently demonstrate a role for heparan sulfates in this process by inhibiting transmission with soluble heparin. This is an interesting study that suggests a potential mechanism by which EV-A71 is able to establish secondary sites of infection. However, I have some questions/comments to be addressed:

Major

1. Did the authors carry out any characterisation of the DC population following differentiation but prior to use in co-culture? DC populations can vary widely between donors and it would be good to include, at minimum, the details of how the DC populations used in co-culture were initially defined.

Minor

1. Line 111: Could the authors clarify the RT reagent used as SupersScript II is not a Promega product.
2. Line 134: Remove extra full stop after RD99 cells.
3. Line 136: Could the authors clarify the target used to determine the DC cell population? In the Methods and Figure, cells are separated on the basis of CD1a expression, but the authors state here that DC populations were distinguished by DC-SIGN.
4. Line 185-187: The authors discuss assessing a role for PSGL-1 in DC-mediated transmission of EV-A71. However, there is no data or details in the results on PSGL-1, which instead focus on DC-SIGN and heparan sulfates. Do the authors have this data to include in the manuscript? If not reference to this should be removed from the discussion for the sake of clarity.
5. Line 286: missing 'c' from co-cultured.
6. Line 311: Check spelling of Ribavirin

Staff Comments:

Preparing Revision Guidelines

Please return the manuscript within 60 days; if you cannot complete the modification within this time period, please contact me. If you do not wish to modify the manuscript and prefer to submit it to another journal, please notify me of your decision immediately so that the manuscript may be formally withdrawn from consideration by Microbiology Spectrum.

Response to the reviewers comments (Spectrum02822-22)

Reviewer 1

Helgers et al. analyzed the potential role of DCs in EV-A71 dissemination. The experiments are well designed. I have some concerns about the technical issues.

1. Ribavirin could not suppress EV-A71 infection (see Smee et al., 2016, Antiviral Research). Guanidine HCl (2 to 4 mM) or rupintrivir (1 μ M) would be useful for the aim. Suppression of EV-A71 infection by these inhibitors must be confirmed to validate the experiments. Therefore, the potential effect of replication in DCs could not be excluded.

As requested, we have now included validation of the inhibitor ribavirin (10 μ M) and rupintrivir (1 μ M) and show that ribavirin blocks replication of EV-A71 C1 and C5 in RD99 cells similar as observed for rupintrivir [Supplementary figure 1; 143; 331 - 335]. Moreover, we have performed infection experiments in DCs, with and without rupintrivir 1 μ M, to show that there is no replication of EV-A71 C1 or C5 in DCs [Supplementary figure 2; 157 – 158; 336 – 342]. These data support our previous results that viral replication does not occur in DCs.

2. Protein secretion is inhibited in enterovirus-infected cells (Spagnolo and Kirkegaard, 2009, JVI). Therefore, a cell surface marker CD1a might not serve as the marker for the EV-A71-infected DCs. To clarify the absence of EV-A71 infection in DCs, the authors should add the data of only DCs, thus in the absence of RD99 cells, in the context of experiments in Fig. 1.

As requested, we have included new data concerning infection of DCs alone with EV-A71 C1 and C5 [Supplementary figure 2; 157 – 158; 336 – 342]. Moreover, we have included a characterization of DC surface markers and CD1a expression is expressed at high levels by DCs [Supplementary figure 1; 71 – 73; 157 – 158; 331 - 335]. CD1a is not a secretory protein but a transmembrane protein expressed on the cell surface as well as intracellular by DCs and not downregulated upon activation. In addition, we have determined intracellular and extracellular CD1a expression to ensure detection of CD1a.

3. L136: CD1a?

We have corrected this mistake.

Reviewer 2

Helgers et al present a study investigating the role of dendritic cells (DC's) in the transmission of EV-A71. Using a combination of flow cytometry and qRT-PCR the authors demonstrate that, while not susceptible to EV-A71 infection themselves, DC's are capable of transmitting infectious virus to susceptible cells following co-culture, and doing so in a replication-independent manner. They subsequently demonstrate a role for heparan sulfates in this process by inhibiting transmission with soluble heparin. This is an interesting study that suggests a potential mechanism by which EV-A71 is able to establish secondary sites of infection. However, I have some questions/comments to be addressed:

Major

1. Did the authors carry out any characterisation of the DC population following differentiation but prior to use in co-culture? DC populations can vary widely between donors and it would be good to include, at minimum, the details of how the DC populations used in co-culture were initially defined.

As requested, we have now included a flow cytometry analysis of DCs including the expression of DC-SIGN, CD1a, CD80, CD83 and CD86 [Supplementary figure 1; 71 – 73; 157 – 158; 331 - 335].

Minor

1. Line 111: Could the authors clarify the RT reagent used as Superscript II is not a Promega product.

We have corrected this mistake. The SuperScript II used is from ThermoFischer Scientific.

2. Line 134: Remove extra full stop after RD99 cells.

We have corrected this mistake.

3. Line 136: Could the authors clarify the target used to determine the DC cell population? In the Methods and Figure, cells are separated on the basis of CD1a expression, but the authors state here that DC populations were distinguished by DC-SIGN.

We have corrected this mistake and clarified that CD1a expression is used to distinguish the DC-population.

4. Line 185-187: The authors discuss assessing a role for PSGL-1 in DC-mediated transmission of EV-A71. However, there is no data or details in the results on PSGL-1, which instead focus on DC-SIGN and heparan sulfates. Do the authors have this data to include in the manuscript? If not reference to this should be removed from the discussion for the sake of clarity.

We have corrected this mistake and removed this section in the discussion.

5. Line 286: missing 'c' from co-cultured.

We have corrected this mistake.

6. Line 311: Check spelling of Ribavirin

We have corrected this mistake.

September 27, 2022

Dr. Adithya Sridhar
Amsterdam UMC
Medical Microbiology
Amsterdam
Netherlands

Re: Spectrum02822-22R1 (Human dendritic cells transmit Enterovirus A71 via heparan sulfates to target cells independent of viral replication)

Dear Dr. Adithya Sridhar:

Your manuscript has been accepted, and I am forwarding it to the ASM Journals Department for publication. You will be notified when your proofs are ready to be viewed.

Congratulations. The ASM Journals program strives for constant improvement in our submission and publication process. Please tell us how we can improve your experience by taking this quick Author Survey.

Sincerely,

Vaithilingaraja Arumugaswami
Editor, Microbiology Spectrum
